# Maternal Supply of Both Arachidonic and Docosahexaenoic Acids Is Required for Optimal Neurodevelopment

**DOI:** 10.3390/nu13062061

**Published:** 2021-06-16

**Authors:** Sanjay Basak, Rahul Mallick, Antara Banerjee, Surajit Pathak, Asim K. Duttaroy

**Affiliations:** 1Molecular Biology Division, ICMR-National Institute of Nutrition, Indian Council of Medical Research, Hyderabad 500 007, India; sba_bioc@yahoo.com; 2A.I. Virtanen Institute for Molecular Sciences, University of Eastern Finland, 70210 Kuopio, Finland; rahul.mallick@uef.fi; 3Department of Medical Biotechnology, Faculty of Allied Health Sciences, Chettinad Academy of Research and Education (CARE), Chettinad Hospital and Research Institute (CHRI), Kelambakkam, Chennai 603 103, India; antara.banerjee27@gmail.com (A.B.); surajit.pathak@gmail.com (S.P.); 4Department of Nutrition, Institute of Basic Medical Sciences, Faculty of Medicine, University of Oslo, 0317 Oslo, Norway

**Keywords:** arachidonic acid,20:4n-6, brain, docosahexaenoic acid,22:6n-3, fetus, maternal diet, cognitive, infants, neurodevelopment, neurogenesis

## Abstract

During the last trimester of gestation and for the first 18 months after birth, both docosahexaenoic acid,22:6n-3 (DHA) and arachidonic acid,20:4n-6 (ARA) are preferentially deposited within the cerebral cortex at a rapid rate. Although the structural and functional roles of DHA in brain development are well investigated, similar roles of ARA are not well documented. The mode of action of these two fatty acids and their derivatives at different structural–functional roles and their levels in the gene expression and signaling pathways of the brain have been continuously emanating. In addition to DHA, the importance of ARA has been much discussed in recent years for fetal and postnatal brain development and the maternal supply of ARA and DHA. These fatty acids are also involved in various brain developmental processes; however, their mechanistic cross talks are not clearly known yet. This review describes the importance of ARA, in addition to DHA, in supporting the optimal brain development and growth and functional roles in the brain.

## 1. Introduction

The neurodevelopmental process involves a complex interplay between nutrients, genes, and environmental factors that result in the optimal growth, development, and maturation of the brain. The development of the brain in utero critically depends on the maternal supply of several components for its well-regulated structural–developmental process, characterized by specifically defined developmental periods, growth, a cellular signaling system, and maturation. During the brain growth spurt, neurodevelopment is particularly vulnerable to nutritional deficiencies [1].

The long-chain polyunsaturated fatty acids (LCPUFAs), docosahexaenoic acid,22:6n-3 (DHA), and arachidonic acid,20:4n-6 (ARA) are important nutrients required for fetal brain growth and development. The accumulation of DHA and ARA in the fetal brain predominantly occurs in the third trimester of a human pregnancy. The de novo synthesis of these LCPUFAs seems low in a growing fetus and placenta [2]; the maternal intake of these fatty acids contributes a significant share for brain development. Maternal DHA and ARA are accumulated rapidly within the cerebral cortex during the last trimester of pregnancy and postnatal 18 months [2,3,4]. The dietary balance of DHA and ARA intake and their interactions are thought to be important for the development and function of the brain. Several experimental studies suggested a crucial involvement of these two fatty acids in neural membrane formation and various roles of their metabolites, production of eicosanoids, and their influence on depression- and anxiety-related behaviors. Moreover, multiple trials have found that higher plasma or erythrocyte DHA levels positively correlate with infant neurocognitive outcomes [5,6,7,8,9,10]. 

High levels of DHA in the brain are achieved during early life and are maintained throughout life. DHA accretion to the brain continues into childhood, and the incorporation of DHA is still high despite its reduced accumulation rate. The preferential transfer of maternal DHA and ARA by the placenta to fetal circulation and its mechanisms are reviewed extensively [11,12]. LCPUFA content in human milk also regulate the amount of DHA and ARA transferred to the infant during breastfeeding [13]. However, this phenomenon may depend on an optimal maternal dietary intake of DHA from the supplement or marine fish [14], whereas ARA levels are usually maintained due to high n-6/n-3 ratios in the diet. 

In the central nervous system (CNS), the proportion of DHA with other membrane fatty acids increases as the brain size increases. The increase in the proportion of these fatty acids continues for the second year of life. DHA has significant neurobiochemical roles in ion channel and receptor functions, the release of neurotransmitters, synaptic plasticity, and gene expression in the neurons. Both DHA and ARA of synaptic membrane phospholipids are released as free fatty acids (FFAs) by activated phospholipase A_2_ (PLA_2_) and are converted to different bioactive metabolites. These liberated fatty acids and the metabolites play critical roles during ischemia seizure activity, inflammation, and other types of brain disorders. Figure 1 describes the putative roles of DHA in neural membranes.

Maternal n-3 PUFA deficiency during pregnancy was associated with impaired brain development in offspring [15] and defective neuroblast migration [16]. The deficiency of n-3 LCPUFAs during development causes hypomyelination in the brain, resulting in mood and anxiety disorders [17,18]. Consuming a diet containing a high amount of n-3 fatty acids during pregnancy protected infants against the detrimental effects of maternal stress [19]. 

The essentiality of DHA is well recognized in childhood and adult life, as its deficiency causes cognitive decline and other psychiatric disorders [20]. Plasma DHA levels are inversely correlated with depressive symptoms in infants and adolescents with bipolar disorder [21]. Human breast milk containing higher n-3 and n-6 LCPUFAs was associated with decreased infant despair and distress [22]. The impact of n-3 PUFAs in human milk in influencing infant mood or anxiety is still not clearly established, since cortisol levels in milk are also associated with infant temperament [23]. Moreover, the variation in the fatty acid composition of mothers’ milk may play an essential role in the outcome of offspring’s mental status and overall health. 

Although n-3 LCPUFAs supplementation is beneficial in preventing and treating major depression, bipolar disorder, and anxiety disorders in adults [24,25], much less is known about how the imbalance of these LCPUFA levels impact the mood and behavior of infants. Studies in experimental models suggested that early exposure to n-3 fatty acids had a lasting effect on temperament and behavioral phenotypes of offspring [26]. Interventional studies in adults also showed an association between the n-3 PUFA status with improved mood and mental health [17,26]. 

Though precise molecular mechanisms are not well defined, DHA and its bioactive derivatives play various essential structural and functional roles in the brain [12,27,28]. High DHA levels in the phospholipids of synaptic membranes provide membrane flexibility and improve the efficiency of protein–protein interaction necessary for signal transduction [27]. Different aspects of metabolism, and the structure–function of the brain depend on optimum ARA and DHA levels and interactions between their metabolites [29]. Cognitive benefits from supplementation of combined ARA and DHA in early life have also been observed through early and middle childhood [30,31]. Furthermore, several human brain diseases, such as Alzheimer’s and bipolar disorders, involve disturbed n-3 and n-6 LCPUFA uptake and metabolism. Therefore, understanding the dynamics of the maternal supply of DHA and ARA to the developing brain may be important for managing brain disorders. An additional neuroprotective mechanism may involve bioactive molecules derived from DHA and ARA, which are also involved in several cellular neuronal biochemical processes. These bioactive derivatives modify the functions of several genes in the brain by acting as ligands for transcription factors involved in critical brain functions, including signal transduction and synaptic plasticity. 

A literature search was performed on the PubMed database by using search terms such as DHA, ARA, brain development, infant, fetal, breastfeeding nutrients deficiency, DHA and ARA supplementation. All types of articles related to human and mechanistic studies on models were included for evaluation. The articles for which full text was not available or not reported in English were excluded. The articles retrieved in the first round of searches identified additional references by a manual search among the cited references. This review describes the latest development of the interplay of DHA and ARA transfer and their impacts on brain development, their complementarity, the structure–function relationship, and their mechanisms of action in the brain. Moreover, the evidence of the essentiality of ARA in brain development is summarized. 

## 2. Maternal Delivery of DHA and ARA to the Developing Brain

Both DHA and ARA are major components of the brain. DHA comprises 10–20% of the total fatty acid composition in the brain, whereas 9% is present in the form of ARA [32]. During the third trimester of pregnancy, these LCPUFAs preferentially accumulate in the fetus and reach higher fetal/neonatal blood levels than those in the mother [33]. Both DHA and ARA are deposited in large amounts relative to the accretion rates of other fatty acids in the fetal brain during a maximum brain growth spurt, which occurs from the last trimester in utero and continues through the breastfed postnatal life [4,34,35,36]. The development of the brain is critically dependent on the adequate maternal supply of LCPUFAs in this period, since their synthesis from the parent EFAs is insufficient to meet the high requirement [4,37]. The lower fetal status of both DHA and ARA is associated with neurological development [38]. Consequently, ample fetal and neonatal LCPUFA supply via transplacental transport and human milk is critically important, implying that maternal LCPUFA status should be adequate. 

Maternal dietary intakes of fatty acids influence the fatty acid composition of breast milk and plasma levels of lactating women and their infants [39]. DHA incorporation in the neuronal membrane in early fetal life solely depends on placental transfer [11], breastfeeding, and the endogenous synthesis of DHA [40,41,42]. However, DHA accretion in the CNS depends on the dietary provision of DHA, i.e., on the duration and concentration of DHA supplementation. A similar nutritional dependency is absent for ARA accretion in the brain, i.e., the dosage and duration of postnatal ARA supplementation do not affect ARA accretion in the CNS [43]. 

A robust linear relationship between maternal DHA level and umbilical cord blood phospholipid was reported [44]. Both ARA and DHA also have a role in the early placentation process, in addition to their roles in fetal neurodevelopment and in the postnatal lactation period [45,46,47]. The high-affinity placental plasma membrane, fatty acid-binding protein (p-FABPpm), is involved in the preferential supply of both maternal DHA and ARA to the fetus [12,41,48]. Since the rapid deposition of these LCPUFAs into the brain occurs during the last trimester of pregnancy and subsequently in lactation, the maternal status must be maintained well during the critical time of brain development. The dietary intake and maternal stores of DHA are the determinants of infant blood DHA concentrations at birth [49]. Blood LCPUFAs in breast-fed infants remain higher than those in maternal circulation postnatally [50,51]. The blood levels of PUFAs of infants are higher than those in the maternal circulation postnatally during the breastfeeding period. The preferential postnatal deposition of LCPUFAs in the infant’s brain is mediated via breastmilk. The dietary supplementation of DHA to pregnant and nursing mothers dose-dependently increases the DHA level in breast milk, which causes higher tissue accretion of DHA in breastfed infants with improved outcomes of mental performance [52,53,54]. The erythrocyte DHA status of breastfed infants is correlated with the maternal DHA status of erythrocytes during lactation; however, no such association was observed for ARA [13]. Human breast milk levels of DHA and ARA are relatively stable throughout the lactation [55]. The worldwide mean concentration of DHA is 0.3  ±  0.2%, and that of ARA is 0.5  ±  0.1% in breast milk [56]. Typically, human breast milk has 1.5- to 2-fold more ARA than DHA, though the breast milk’s DHA content can be higher than ARA in populations with high marine fish consumption. It has been shown that the DHA level in breast milk is directly related to the DHA content of the maternal diet [57]. Still, it is unknown whether metabolic or dietary mechanisms explain the lower variability in breast milk ARA. Breast milk ARA was not affected in lactating mothers allocated to consume increasing doses of DHA [57]. Two recent studies confirmed the different regulation of ARA and DHA in breast milk, indicating that ARA is affected by the genetic pattern in the *FADS*-gene cluster [58] and is less sensitive than DHA to dietary supplementation [59]. The essentiality of ARA in infant nutrition is supported by the observation of the potentially adverse effects in preterm infants of consuming a marine-oil-containing formula [60]. Dietary ARA has roles in growth, neuronal development, and cognitive function in infants. Both ARA and DHA are necessary for fetal development, and a deficiency in one may compromise growth [61]. It has been shown that growth deficiency induced by fish oil supplementation is related to a reduced ARA availability due to the excess of DHA [62]. Western diets usually have a much higher ratio of n-6/n-3 fatty acids [63]. Therefore, it has been proposed that supplementation during gestation should be based on intake of n-3 fatty acids.

Children’s intelligence quotient (IQ) was increased by 0.8 to 1.8 points when their mothers consumed DHA from pregnancy to the lactation period and beyond [64,65]. The mean levels of DHA and ARA in breast milk are found at 0.37% and 0.55% of total fatty acids across the globe, respectively. Prospective observational studies suggested that breastfed infants had a significant neurocognitive advantage compared with formula-fed infants [5,66,67], possibly due to the higher incorporation of DHA and ARA in breast milk relative to formula milk. The association between breastfeeding and child IQ concerning FADS2 genetic profile, specifically in SNP rs174575b, was observed [68]. Breastfed infants with rs174575 C-dominant carriers achieved higher scores on standardized IQ tests than non-breastfed C-carrier infants. However, observational data are confounded by the heterogeneous composition of breast milk, and environmental factors that influence the infant’s mental development.

Based on established guidelines, it is emphasized that maternal dietary DHA requirements should be increased during pregnancy and lactation. Precisely, a minimum of 200 mg of DHA per day is recommended during these periods [69]. In both full-term and preterm populations, the evidence is compelling that breastfeeding is vital for an infant’s neurodevelopment. 

## 3. The Fatty Acid Uptake System of the Brain

The de novo synthesis of DHA in the brain is almost non-existent, and therefore, it must be imported from the circulation across the blood–brain barrier (BBB) [70]. The BBB comprises endothelial cells in the capillary connected with tight junctions, astrocytic end-foot processes, pericytes, and neurons [71]. The endothelial cells offer highly selective permeability in the BBB by their specialized tight junction function. Thus, the BBB is provided with a neuronal system immune-privileged environment permitting only small molecules into the brain. 

Free fatty acids (FFAs) are transported into the cytosol via cell membrane- and cytoplasmic fatty acid-binding proteins (FABPs) [12]. There are four classes of membrane fatty acid transport proteins present in the brain, such as fatty acid translocase (FAT/CD36), plasma membrane fatty acid-binding proteins (FABPpm), fatty acid transport proteins (FATPs), and several cytosolic fatty acid-binding proteins (FABPs) [41,72,73]. Additionally, another transporter, Mfsd2a (major facilitator superfamily domain-containing protein 2A), is present in the BBB [12]. Mfsd2a transports lysophosphatidylcholine (LPC)-DHA, but not unesterified DHA [12,74,75,76]. The DHA-LPC produced from DHA-containing phosphatidylcholine when acted upon by PLA1. A sterol regulatory element-binding protein regulates the activity of Mfsd2a to maintain a balance between de novo lipogenesis and exogenous uptake of LPC-DHA. The brain of the Mfsd2a-deficient mice had significantly reduced DHA levels and experienced loss of neurons in the hippocampus and cerebellum. The mice later developed microcephaly with severe cognitive deficits and anxiety. Altered plasma levels of Mfsd2a during pregnancy influence the placental transport of DHA and neurodevelopment in utero [77]. The maternal blood levels of Mfsd2a in the third trimester were inversely correlated to DHA-LPC in maternal plasma [77]. DHA or DHA-LPC is taken up by the endothelial cells via FAT/CD36, FATPs, or Mfsd2a. Plasma albumin binds both unesterified DHA and DHA-LPC. 

Like LPC-DHA, the accretion of ARA to the brain from ARA-containing lysophospholipids is six-fold more efficient than for unesterified ARA [78]. The brain relies on different transport systems for shuttling the varied forms of the metabolized DHA across the BBB. Active pathways of DHA and ARA incorporation into the brain include members of the FABP family. In human brain microvessel endothelial cells, FAT/CD36 and FATP-1, FATP4 also transport fatty acids across the monolayer [73]. However, DHA incorporation into brain phospholipids was not affected in CD36_−/−_ mice, indicating this transporter may not be involved in the transport of DHA. 

FABPs involve in the FFA uptake and transport to various intracellular compartments of a cell [73,79]. The expression of various FABP genes occurs at different developmental stages of the brain, as shown in animal studies. FABP3 is expressed in the brain after birth, and its expression levels increases until adulthood [80]. FABP4 is mainly expressed in grade IV astrocytomas and normal brain tissue [81], whereas FABP5 is expressed in the mid-term embryonic rat brain and reaches its peak at birth, then gradually decreases in postnatal life [80]. FABP7 is expressed mainly in radial glial cells at the early stages of brain development [80,82]. The level of FABP7 expression decreases starkly in the neonate and adult brain [80]. FABP7 plays a role in the establishment of the radial glial fiber system [82]. Moreover, FABP7 has been suggested to be associated with the decreased survival of glioblastoma patients [83,84,85]. 

FABPs have broad binding specificity, including the binding affinity for long-chain fatty acids (≥c16), eicosanoids, bile salts, and PPARs [41,86]. Both FABP3 and FABP4 bind ARA with high affinity [87,88], while FABP5 preferentially binds long-chain saturated fatty acids (c ≥ 16) [89]. FABP7 binds both DHA and ARA but with four-fold more affinity for DHA [90], indicating that DHA may be a preferred ligand for FABP7. Although FABP7, FABP5, and FABP3 can also bind different types of fatty acids [89,91]. 

FABP3, FABP5, and FABP7 are involved in both developing and mature adult brains [86,92]. Functional studies have demonstrated a variety of roles for FABPs in brain development, including the generation of neuronal and/or glial cells, differentiation, neuronal cell migration, and axis patterning. Like DHA, FABP7 increases the proliferation of neural stem cells and neural progenitor cells and differentiates into mature neurons both in vitro and in vivo [93,94]. 

FABPs are multi-functional proteins, and complex signaling networks and transcription factors regulate their expression. FABPs are major downstream effectors of the Reelin-Dab1/Notch pathway that involves neuron–glia crosstalk during brain development. Since LCPUFAs and several FABPs are involved in brain development and function, it is important to further elucidate their roles in brain disorders’ pathogenesis. 

As FABPs are involved in developing, establishing, and maintaining the central nervous system, FABPs are implicated in the pathogenesis of Down syndrome. FABP7 is overexpressed in Down syndrome adult [95] and fetal brains [96], whereas FABP3 is significantly decreased in Down syndrome adult brains [95]. Furthermore, FABP7 upregulation correlates with PKNOX1 gene-dosage imbalance in the brains of Down syndrome patients. PKNOX1 is a POU domain protein that may directly control FABP7 expression by interacting with the Pbx/POU binding element of the FABP7 promoter [96]. Human FABP7 mRNA levels were significantly upregulated in the postmortem brains of schizophrenia patients. A correlation between an SNP variant within the second exon of human FABP7 and schizophrenia pathology was observed [97]. The impairment of prepulse inhibition (PPI) occurs in many brain disorders such as Alzheimer’s disease, autism, bipolar disorders, Tourette syndrome, and schizophrenia [98]. The association between FABP7 and PPI status suggests roles of the FABP7 gene in the pathology of PPI-mediated neuropsychiatric and/or neurodegenerative diseases. The association of FABP3 and FABP5 with several neurodegenerative diseases is also reported [99,100]. DHA influences brain functions and protects from different brain tumors such as astrocytoma and glioma by binding to the FABP, resulting in the activation of transcription factor PPARγ to the nucleus-reduced cell migration, growth arrest, and apoptosis of tumor cells [101]. However, the roles of these proteins in human brain development are not well known. 

## 4. Structural and Functional Roles of DHA in the Human Brain

The brain is known as the body’s fattiest organ, containing phospholipids around 2/3 of its weight. The brain harness 20% of its total energy from β-oxidation of fatty acids in the mitochondria of astrocytes. The presence of DHA on the membrane influences neuronal information transmission, signal transduction velocity, and interaction with ion channels or receptor proteins and their activity [102]. DHA is also predominantly present in cortical gray matter, representing approximately 15% of total fatty acids in the adult human prefrontal cortex. As a crucial structural ingredient of the brain, DHA comprises the regions of the cerebral cortex and synaptic membrane. Neuronal membranes have approximately 50% DHA [44]. DHA is also vital for hippocampal and cortical neurogenesis, neuronal migration, and outgrowth [93,103,104]. 

The brain fatty acid levels, mostly LCPUFAs, are maintained via different mechanisms, as described above. The circulating plasma levels of DHA is positively related to cognitive abilities during aging and is inversely associated with decline in cognitive function. DHA, being a part of the cell membrane phospholipid, contributes to maintaining optimal fluidity and lipid raft assembly in the membranes, membrane electrical and antigenic signals of the cells. DHA also halts cell death by stimulating cell-cycle exit in neuro-progenitor cells [93,105]. DHA is involved in monoaminergic and cholinergic systems during brain development processes [52,106,107]. DHA has a long-term effect on serotonergic and dopaminergic systems during the fetal brain development in utero [35,107,108]. Data emphasized the importance of examining the long-term critical impact on brain development due to inadequate DHA supplies to the fetus during pregnancy. DHA stimulates neurite outgrowth in cell culture systems. Neurite outgrowth is an important process in the developing nervous system and also in the regeneration of nerves. The alpha linoleneic acid, 18:3n-3 (the precursor of DHA)-restricted diet decreased neurogenesis in rat dams’ fetal brains, possibly due to the deficiency of DHA [103]. DHA influences gene expression, neurotransmission and protects the brain from oxidative stress during development [109]. DHA is an essential factor for neurogenesis, phospholipid synthesis, and turnover [93,110,111]. 

Again, DHA can act as a ligand for peroxisome proliferator-activated receptor-gamma (PPARγ) and retinoid X receptor (RXR). RXR plays a vital role in embryonic neurogenesis, neuronal plasticity, and catecholaminergic neuron differentiation along with retinoic acid receptors. RXR is highly expressed in the hippocampus [112,113]. The PPARγ-RXR heterodimer modulates early brain development by regulating transcription genes [112,114]. 

DHA also protects the developing brain from peroxidative damage of lipids and proteins [115,116,117]. DHA and eicosapentaenoic acid,20:5n-3 (EPA) were reported as suppressors of angiogenesis in cancer cells, but they stimulate angiogenesis in the placenta [45]. However, DHA and DHA-LPC may act as pro-angiogenic and anti-angiogenic depending on the concentration and microenvironments [118]. 

Several epidemiological data show an inverse association of low habitual dietary intake of DHA and a higher risk of brain diseases [2,119,120]. A diet containing high amounts of n-3 fats and/or a lower amount of n-6 fats was strongly associated with the lower incidence of Alzheimer’s disease and other brain diseases [121,122,123,124]. Intake of DHA improves attention deficit hyperactivity disorder (ADHD), bipolar disorder, schizophrenia, impulsive behavior, and other brain disorders [20,123,125].

Nevertheless, the data of intervention studies with DHA supplements are conflicting, despite the fact that many such studies demonstrated an apparent benefit of DHA intake in brain function. Several studies failed to reproducibly show that the absence of DHA and its metabolites are involved in various adult brain diseases. More well-designed clinical trials considering background diets and genetic makeup are needed for definitive conclusions.

## 5. Roles of DHA and Its Metabolites in the Brain

DHA and its metabolites play vital roles in the functional brain development of the fetus in utero and infants and healthy brain function in adults. DHA and its metabolites play significant roles in cellular and biological functions. The oxidation of DHA by lipoxygenases produces several types of metabolites such as oxylipins that regulate various biochemical processes of the brain [126].

DHA stimulates membrane-associated G-protein-coupled receptor (GPR) 120 mediated gene activation to promote anti-inflammatory activities [127,128]. DHA also activates PPARs and upregulates the expression of genes responsible for increasing insulin sensitivity and reducing plasma triglyceride level and inflammation. [44]. DHA and its metabolites’ signaling pathways are involved in neurogenesis, anti-nociceptive effects, anti-apoptotic effects, the plasticity of the synapse, Ca^2+^ homeostasis in the brain, and nigrostriatal activities [129]. DHA itself and its metabolites have a broad spectrum of actions at different levels and sites in the brain [129,130]. Figure 2 shows the DHA and EPA metabolites and their function in the brain. 

DHA is converted to maresin 1(MaR1), neuroprotectin D1(NPD1), and resolvins by human 12-LOX, 15-LOX, and CYP or enzymatically by aspirin-treated COX-2. They play various functions in the brain. DHA is metabolized by the P450 system, cyclooxygenase, and lipoxygenase enzymes under different metabolic conditions. 14-HDHA: 14-hydroxy-docosahexaenoic acid; 17-HDHA: 17-hydroxy-docosahexaenoic acid; 18-HEPE: 18-hydroxy-eicosapentaenoic acid; LG R6: G protein-coupled receptor 6; ALX/Fpr2: N-formyl peptide receptor 2; BLT1: leukotriene B4 receptor; COX-2: cyclooxygenases; DHA: docosahexaenoic acid; EPA: eicosapentaenoic acid; GPR32/37: G protein-coupled receptor 32/37; LC-PUFAs: long-chain polyunsaturated fatty acids; LOX: lipoxygenases.

DHA-derived specialized pro-resolving mediators (SPMs) such as DHA epoxides, oxo-derivatives (EFOX) of DHA, neuroprostanes, ethanolamines, acylglycerols, docosahexaenoyl amides of amino acids, and branched DHA esters of hydroxy fatty acids play important roles in brain functions [131,132]. Additionally, epoxydocosapentaenoic acids (EDPs) and 22-hydroxydocosahexaenoic acids (22-HDoHEs) are produced from DHA [133,134]. DHA is mainly metabolized by enzymes such as 5-, 12- and 15-lipoxygenases (LOX), COX-2, and cytochrome P450 (CYP). As the most demanding by-products of DHA, resolvins are formed by either LOX15 or CYP or aspirin-treated COX-2 activity [28]. The LOX15-derived resolvins are homologous to CYP-, or aspirin-treated COX-2-derived resolvins [28]. As the inflammation resolution mediators, resolvins act via different G-protein coupled receptors (GPRs) [28,135]. Both resolvin D1 and aspirin-triggered resolvin D1 improve brain functions and impede neuronal death by down-regulating several factors such as NFkB, TLR4, CD200, and IL6R [136,137]. They even induce remote functional recovery after brain damage [136]. Both resolvin D2 and aspirin-triggered resolvin D2 protect from cerebral ischemic injury via phosphorylation ERK1/2. Subsequently, this pathway stimulates nNOS or eNOS to inhibit neuronal cell death and maintain BBB integrity by increasing zonula occludens-1 [138]. Resolvin D3, resolvin D5, aspirin-triggered COX-2 -derived resolvin D3, and aspirin-triggered resolvin D5 halt the neuroinflammation [139,140]. However, the functions of other resolvins are still a mystery. Maresin (MaR), the anti-inflammatory pro-resolving mediator, is produced from DHA in macrophages during the inflammation, healing, and regeneration process [141,142,143]. MaR1 is predominant among other maresins. MaR1 decreases LTB_4_ synthesis and stimulates phagocytosis at the site of inflammation [144,145,146]. MaR1 enhances tissue repair by stimulating stem cell differentiation and plays an analgesic role through TRPV1-mediated response blockage [145,147]. MaR1 involves neurocognitive functions by regulating the infiltration of macrophages, regulating NF-κB signaling, oxidative stress, and cytokine release. Maresin 1 reduces neuroinflammation perioperative neurodegenerative disorders in an animal model [148]. MaR1 significantly affects the post-spinal cord injury model [149,150]. 

Another SPM, neuroprotectin D1(NPD1), derived from DHA, improves cell survival and cell repair in brain disorders [151]. Like MaR1, NPD1 also possesses anti-inflammatory and neuroprotective activities [152]. In response to neuroinflammation, NPD1 is produced from endogenous DHA in the retina and brain [153,154]. Besides antiviral protection, NPD1 helps in neurocognitive functions [155,156,157]. NDP1 blocks the progression of Alzheimer’s disease by stimulating the expression of PPARγ, amyloid precursor protein-α, and reducing the β-amyloid precursor protein [156]. Figure 3 describes DHA metabolites and their global effects on gene expression and second messenger systems affecting multiple cellular functions in the brain.

Maresin 1(MaR1), neuroprotectin D1(NPD1), and resolvins are produced from DHA by human 12-LOX, 15-LOX, and CYP or aspirin-treated COX-2 enzymatically. These metabolites have multiple functions in the brain, which have been mentioned in the boxes. DHA is metabolized by the P450 system, cyclooxygenase, and lipoxygenase enzymes under different metabolic conditions. 

Anti-inflammatory activities of EFOX and neuroprostanes protect neuroinflammation in various diseases, such as Parkinson’s disease and Alzheimer’s disease [158,159,160]. Another DHA derivative, docosahexaenoyl ethanolamide, improves mood, pain, inflammation status, hunger, and glucose uptake by the brain endocannabinoid system [161,162,163,164,165,166]. The function of DHA metabolites is summarized in Table 1.

DHA glyceryl ester regulates the intake of food and neuroinflammation, similarly to the way docosahexaenoyl ethanolamide uses the endocannabinoid system [167,168]. The endocannabinoid system plays an integral part in memory, cognition, and pain perception [169,170]. DHA conjugates via cannabinoid receptors reduce neuroinflammation and improve neurogenesis [171,172]. 

DHA is involved in alleviating short-term stress, preventing anxiety and stress in later life [132,173]. DHA is reported to improve various psychiatric disorders such as schizophrenia, mood and anxiety disorders, obsessive-compulsive disorder, ADHD, autism, aggression, hostility and impulsivity, borderline personality disorder, substance abuse, and anorexia nervosa [174]. There is strong evidence that the consumption of marine fish reduces depression [175]. 

The mild symptoms of ADHD are corrected with DHA supplementation [176]. DHA was also shown to improve depressive symptoms of bipolar disorder by increasing N-acetyl-aspartate brain levels without affecting mania [174]. Even IQ outcomes in children and cognitive function in the aging brain are improved by DHA supplementation [2,24]. DHA-derived anti-inflammatory eicosanoids’ neuroprotection prevent Alzheimer’s disease pathogenesis [177]. DHA may modulate the metabolism of cholesterol and apolipoprotein E, lipid raft assembly, and the cell signaling system in Alzheimer’s disease [178,179]. DHA protects neuronal brain function by reducing NO production, calcium influx, and apoptosis while activating antioxidant enzymes such as glutathione peroxidase and glutathione reductase [180]. The second most prevalent neurodegenerative disease, Parkinson’s disease, can be halted by DHA’s neuroprotective role [180]. 

## 6. DHA Deficiency in Utero and Human Brain Function

DHA deficiency is linked with different brain disorders such as major depressive and bipolar disorder [181,182]. DHA levels are positively correlated with improved learning and memory and reduced neuronal loss [24]. DHA deficiency affects epigenetic development in the feto-placental unit [183,184]. The improvement in attention scores, adaptability to new surroundings, mental development, memory performance, and hand–eye coordination are associated with higher maternal DHA delivery to the fetal brain [7,185]. 

DHA deficiency during pregnancy suggests the lower development of language learning skills in children [186]. Even autistic spectrum disorder or ADHD among teenagers is associated with DHA deficiency [187,188]. Neurocognitive functional insufficiency in young adults or loneliness-related memory problems in middle age have been associated with DHA deficiency [189,190]. DHA deficiency in the third trimester significantly causes preterm brain development due to the insufficient maternal consumption of n-3 fatty acids. Even following delivery, infants are entirely dependent on breast milk or formula milk for DHA and ARA. Reduced DHA consumption during this critical brain development period may influence brain functionalities in adult life [191]. 

Dementia has shown an inverse relationship with regular marine fish consumption in different continents [120,192] and higher blood DHA levels inversely related to dementia [193,194]. Various brain diseases/disorders, such as Alzheimer’s disease, Parkinson’s disease, Huntington’s disease, schizophrenia, and mood disorders, are related to disturbed fatty acid signaling [195,196]. The most prevalent dementia is Alzheimer’s disease, which is inversely related to brain DHA level. Serum DHA level reduces significantly, and its addition positively correlates with memory scores in elderly Alzheimer patients [119,197,198]. The plasma DHA level is significantly associated with the risk of Alzheimer’s disease [199]. Intake of 200 mg of DHA-containing fish per week reduces the risk of AD by 60% [200]. However, different randomized controlled trials (RCTs) found mixed results with DHA supplementation. DHA alone or combined with ARA or the EPA found no significant neuropsychiatric status changes in Alzheimer’s disease patients [123,124,201]. The neurodegenerative disorder, Parkinson’s disease’s etiology is unknown. However, its primary palliative treatment is dopamine-based therapy. Various animal studies found the neuroprotective effect of DHA in the Parkinson’s disease model. DHA has been shown to improve L-DOPA-induced dyskinesia [202] and reduce dopaminergic neuron apoptosis in mouse models [121]. DHA supplements offer a beneficial neuroprotective effect for Parkinson’s disease management [203]. 

Multiple factors influence serotonin biosynthesis and function. The brain’s serotonin level correlates with various behavioral consequences, e.g., control of executive function, sensory gating, social behavior, and impulsivity [204]. Serotonin-related gene polymorphism is associated with mental illnesses, e.g., autism spectrum disorders, ADHD, bipolar disorder, schizophrenia, etc. DHA modulates the activity of serotonin in the brain. DHA increases serotonin receptor accessibility by increasing membrane fluidity in postsynaptic neurons [204]. Concentric serotonin and a low DHA level in the orbitofrontal cortex are correlated with schizophrenia [205]. 

Observational studies showed that ADHD also has a relationship with DHA levels. RCTs of DHA with EPA supplementation and the addition of medications have demonstrated significant improvement in ADHD symptoms [122,187,206,207,208,209,210,211,212,213,214,215,216]. However, DHA supplementation with methylphenidate did not improve ADHD symptoms [217]. 

There are mixed results in RCTs that have been observed in early psychosis symptoms improvement with DHA supplementation. When DHA is supplemented with EPA for at least 12 weeks, the functional improvement and reduction in psychiatric symptoms are visible in different studies [218,219,220]. DHA deficiency elicits the chances of schizophrenia by promoter hypermethylation of nuclear receptor genes RxR and PPAR, which results in the downregulation of the gamma-aminobutyric acid-ergic system and the prefrontal cortex involved in oligodendrocyte integrity [221]. Lower erythrocyte DHA status is associated with the development of bipolar disorder [222]. However, combined DHA and EPA supplementation for 6 weeks improved mania and depression among juvenile patients [223,224,225]. However, recent RCTs found DHA has no role in improving the symptoms of bipolar disorder [20]. 

## 7. Can DHA Supplementation Improve Brain Function of Infants: Results of Clinical Trials 

Various RCTs proved significant effects of DHA supplementation on infant brain development in pregnancy. The meta-analytic study in 2007 found a static co-relation between visual growth and DHA supplementation in first year of life [125]. An RCT in 2011 showed DHA-supplemented 18-month-old children had higher index scores in mental development [8]. Other RCTs showed DHA-enriched fish oil supplemented children had significantly higher percentile ranks of the total number of gestures at 1 to 1.5 years of age [226]. DHA-formula-fed infants scored equal visual equity scores with breast-fed infants at the age of four [227]. The recent meta-analytic review also found the positive effects of seafood consumption in pregnancies in developing childhood neurocognitive function [53]. The supplementation of DHA to pregnant and nursing mothers and the first year of infant life have developed better cognitive ability. During the last trimester of pregnancy, fetal brain development demands a higher amount of DHA, which can be interrupted in the case of a preterm born baby and can result in mental growth retardation. Different RCTs showed that higher DHA-enriched formula (around 1% of total fatty acids) is essential for the preterm baby for mental growth and development [24]. 

DHA supplemented along with EPA enhance the outcome for cognitive and mood disorders. However, conflicting data exist about the effect of DHA supplementation on cognition during childhood. The earlier RCT conducted in Australia and Indonesia did not show any improvement in general intelligence or attention among 6–10-year-old children following 88 mg/d DHA supplementation [228]. The DHA Oxford Learning and Behavior (DOLAB) study showed a significant improvement in brain function among aged 7–9-year-olds [229]. Another RCT showed improved memory and learning ability among 7–9-year-old children [230]. A few years ago, another RCT from Australia found no significant difference in academic performance between the DHA-supplemented and control groups [231]. Although the authors have mentioned their study’s limitation that the final assessment was finished by less than half of the study population, DHA supplementation was low. The study was conducted with slightly older children, and in some cases, mothers did not give consent or provide data. The Third National Health and Nutrition Examination Survey (NHANES III) found a higher DHA supplementation effect in girls than boys, despite both sexes’ receiving cognitive benefit [232]. The pregnant women need higher DHA supplementation for their own growth and for the growth and development of the newborn. Even the source of DHA and the ratio of EPA and DHA may influence the bioavailability of DHA. 

A summary of a few recent clinical trials has been shown in the following Table 2. Although a slight improvement, these studies did not show any significant positive outcome from DHA supplementation effects on young children. However, a meta-analysis of DHA supplementation with the EPA improved childhood visual and psychomotor development without significant global IQ effects later in childhood [233]. However, there are no clinical data available where DHA supplementation was conducted before 14.5 weeks of the gestational period. 

## 8. Roles of Arachidonic acid20:4n-6 (ARA) in Brain Development and Function

In addition to DHA, the mother preferentially supplies ARA to the growing and developing brain via the placenta and breastfeeding. ARA uptake was found to be higher in early trimester trophoblast cells than EPA and DHA [239]. The ARA metabolism in the brain is suspected of having an altered profile in neurological, neurodegenerative, and psychiatric disorders. Using various knock-out models for enzymes involved in brain ARA metabolism, Bosetti showed that the ARA and its metabolites play a significant role in brain physiology via the PLA2/COX pathway [240]. The effects of DHA and ARA on body growth and brain functions were studied using delta-6-desaturase knock-out (D6D-KO) mice by feeding different combinations of PUFAs in milk formulations. The in vivo findings confirmed the complementary roles of ARA and DHA in body and brain development, respectively [241]. ARA may be required in a higher amount to support growth-promoting placental activities and the production of eicosanoids. In human milk, the amount of ARA typically exceeds the levels of DHA. Milk ARA content is also less varied than DHA, and, unlike DHA, ARA does not seem to be linked to maternal intake. There has been much discussion in recent years about the need for ARA and DHA in infant formula. Studies clearly show the requirement for both ARA and DHA in addition to the essential fatty acids (linoleic acid,18:2n-6 (LA), and ALA to support the optimal body, brain growth, and brain function. ARA is quantitatively the most common LCPUFA in the brain after DHA [242]. Although diverse roles of DHA are investigated, the roles of ARA in brain development and functions have not been investigated to a greater extent. Given that ARA and its precursor, LA, contribute significantly to the Western diet and its pleiotropic biological effects and its interactions with DHA, this n-6 LCPUFA is a crucial modifiable factor in brain development and preventive strategies of brain diseases. ARA corresponds to around 20% of the total amount of neuronal fatty acids and is mainly esterified in membrane phospholipids. 

Several studies have suggested that the structure–function and metabolism of the brain depend on levels of ARA and DHA and interactions of their metabolites [29]. The recycling (de-esterification–re-esterification) of these two fatty acids in the brain are independently carried out by ARA- and DHA-selective enzymes. The ARA-mediated processes can be targeted or altered separately from the DHA-mediated processes by a dietary deficiency of n-3 PUFA or genetic manipulation. Therefore, in studies using n-3 PUFA deficiency models, the homeostatic mechanisms show DHA loss in the brain while increasing ARA metabolism. Further studies are required to understand the impact of the n-6/n-3 ratio on the regulation of DHA-selective iPLA2 and COX-1 or ARA-selective cPLA2, sPLA2, and COX-2 and their effects on brain function and neuroinflammation.

ARA must either be consumed in the diet or synthesized from its precursor LA in the liver. The brain contains relatively low LA levels, and its conversion into ARA is minimum in the brain. Thus, the growing brain depends on a steady supply of ARA [243] from the maternal circulation or via breast milk. Although lipoproteins and lysophospholipids of plasma may contribute to brain ARA levels, their quantitative contribution is unknown. Upon its entry into the brain, ARA is activated by a long-chain acyl-CoA synthetase and can be esterified into the sn-2 position of phospholipids. During neurotransmission, the brain ARA cascade is initiated when ARA is released from synaptic membrane phospholipid by the neuroreceptor-initiated activation of cPLA_2_. PLA_2_ is activated by dopaminergic, cholinergic, glutamatergic, and serotonergic stimulation via G-proteins or calcium [244]. Several PLA_2_ are activated via serotonergic (5-hydroxytryptaminergic), glutamatergic, dopaminergic, and cholinergic receptors [244,245]. Usually, calcium-dependent cytosolic PLA_2_ (cPLA_2_) resides at the postsynaptic terminals, selective for releasing ARA, whereas calcium-independent PLA_2_ is believed to release the DHA sn-2 position of phospholipids [13,14]. Upon its release, a portion of the unesterified ARA is converted to prostaglandins, leukotrienes, and lipoxins, a portion oxidized via β-oxidation, and the remainder (approximately 97% under basal conditions) is activated by ACSL and ultimately recycled and re-esterified into the sn-2 position of phospholipids [246]. An additional ARA is released by activated cytokine and glutamatergic N-methyl-d-aspartate receptors in conditions such as neuroinflammation and excitotoxicity. Although the signals that ARA and its derivatives relay are not entirely understood, they regulate blood flow, neuroinflammation, excitotoxicity, the sleep/wake cycle, and neurogenesis [247].

Like DHA, ARA is also directly involved in synaptic functions. The level of intracellular free ARA and the balance between the releasing and incorporating enzymes in membrane phospholipids may play critical roles in neuroinflammation and synaptic dysfunction. Both these events are observed in the murine model of Alzheimer’s disease before the amyloid plaques and the neurofibrillary tangles, respectively formed by the two agents known for Alzheimer’s disease agents, A β peptide and hyperphosphorylated *tau.* Finally, western food, which contains excessive n-6/n-3 ratios, might favor more ARA levels and influence Alzheimer’s disease mechanisms. 

A better understanding of the complex relationships between ARA and DHA and their brain mechanisms is required. Free ARA contributes to Alzheimer’s disease progression via different pathways. ARA and derivatives are pro-inflammatory and participate in neuroinflammation. ARA is directly involved in synaptic functions as a retrograde messenger and a regulator of neuro mediator exocytosis. ARA also influences tau phosphorylation, and polymerization can compete with DHA. Moreover, ARA has pleiotropic effects on brain disease, and it may be used in the fight against brain diseases. The dietary ARA and brain diseases about DHA should be investigated further to prevent the disease. 

The transgenic Alzheimer’s disease murine model showed that dietary ARA produced opposite Aβ production in Alzheimer’s disease. Studies on the impact of dietary ARA on Alzheimer’s disease are required to identify the underlying mechanisms of action. A reduced level of ARA in the temporal cortex of Alzheimer’s patients was observed [248]; however, its relation with DHA and its metabolites is unknown. ARA involvement in Alzheimer’s disease was mediated via cPLA_2_α. However, a DHA-rich diet did not show such effects [249]. A diet containing 2% ARA for 21 weeks increased Aβ_1-42_ production and deposition in 24-week-old CRND8 mice [250]. 

ARA is involved in cell division and signaling during brain growth and development [251]. In addition, ARA mediates neuronal firing [252], signaling [253] and long-term potentiation [254]. The absolute levels of n-3 PUFAs and the ratio of n-6 and n-3 PUFA affect gene expression of controlling neurogenesis and neural function. 

ARA maintains structural order of the membrane and hippocampal plasticity [255]. ARA also protects the brain against oxidative stress in the hippocampus region via PPARγ and synthesizes new proteins [256]. Released intracellular ARA activates protein kinases and ion channels, inhibits the uptake of neurotransmitters, and enhances synaptic transmission, and modulates neuronal excitability [251]. As ARA is involved in intracellular signaling, the optimum levels of intracellular free ARA must be steadily maintained. ARA also activates syntaxin-3 (STX-3), a plasma membrane protein involved in the growth and repair of neurites [257]. Neurite growth closely correlates with the ARA-mediated activation of STX-3 in membrane expansion at growth cones [257]. The neurite growth from the cell body is a critical step in neuronal development. ARA stimulates exocytosis by allowing the attachment of STX-3 with the fusogenic soluble N-ethylmaleimide-sensitive factor receptors (SNARE complex) [258]. The SNARE proteins are involved in producing a fusion of vesicular and plasma membranes in the brain. The formation of this SNARE complex mediated by ARA drives membrane fusion, leading to the release of vesicular cargo into the extracellular spaces [258]. α-Synuclein plays a role in the development of Parkinson’s disease, can sequester ARA and thus blocks the activation of the SNARE complex [258], suggesting the importance of ARA in synaptic transmission. All these data show the importance of ARA in cell signaling, trafficking, and the regulation of spatial–temporal interactions between cellular structures. 

The most abundant prostaglandins in the brain are PGD_2_ and PGE_2_. They are synthesized from ARA by PGD_2_ and PGE_2_ synthases. COX-2 is overexpressed in the cortex and hippocampus of Alzheimer’s disease patients [259]. PGE_2_ increases neuroinflammation and amplifies Alzheimer’s disease pathology through various mechanisms. 

Figure 4 describes different metabolites of ARA involved in brain function. ARA-derived lipoxins are anti-inflammatory eicosanoids distinct from pro-inflammatory leukotriene and prostaglandin. Lipoxin biosynthesis occurs via two different pathways. Lipoxins mediate their action on endothelial cells to offer an inflammation resolution process. LipoxinA4 lowers neuroinflammation by reducing microglial activation. 5- Lipoxygenase (5-LOX) converts ARA into 5-HPETE and then 5-HETE or leukotriene A4 (LTA4). There is the increased expression of the dual enzyme 12/15-LOX and its products. 

ARA is converted to lipoxinA4 or lipoxinB4 via two different pathways. Lipoxins mediate their action on endothelial cells to offer an inflammation resolution process. LipoxinA4 lowers neuroinflammation by reducing microglial activation. LTA4, LTB4, LTC4, LTD4, and LTE4 are leukotrienes A4, B4, C4, D4, and E4, respectively; DP1, EP1-EP4, FP, and IP, prostaglandin receptors; TP, thromboxane A2 receptor; BLT1 and BLT2, leukotriene B4 receptor; CysLT1 and CysLT2: cysteinyl leukotriene receptors; ALXR: lipoxins receptor.

12(S)-HETE and 15(S)-HETE were reported in frontal and temporal brain regions’ cerebrospinal fluid, respectively, in Alzheimer’s disease patients. 

The endocannabinoids containing ARA are involved in Alzheimer’s disease via the CB2 receptor. Although neuroinflammation is related to Alzheimer’s disease, there is no evidence that higher brain contents of ARA would produce inflammation.

Several animal studies indicate that ARA has beneficial effects on cognition and synaptic plasticity [260]. An association between abilities of spatial memory and ARA content of the hippocampus was reported. ARA modulates Kv channels at the postsynaptic membrane, and influences long-term potentiation [261]. ARA induces presynaptic long-term depression associated with a Ca^2+^ influx and the activation of metabotropic glutamate receptors [261]. In addition, ARA can induce neurotransmitter exocytosis in the presynaptic neuron via activation of the soluble N-ethylmaleimide-attachment receptors (SNARE). ARA induces the binding of syntaxin-1 to the SNARE complex in the presence of Munc18-1, which is a critical regulator of the process [262]. Although ARA’s role in synaptic function is well documented, the benefit or drawback may depend on the delicate balance of ARA levels. Age and Aβ concentration or other physiological factors could modify ARA’s effects on synapse and neuronal cells. Additional work is now required to characterize dietary ARA’s influence on associated brain diseases.

## 9. Transport of ARA to the Developing Brain

The transport of ARA to the brain tissue and inside the brain are not well understood, and more investigations are required for their nuanced characterization. The biological roles of several ACSLs and LPATs for incorporating ARA in the membrane phospholipids have been studied [246]. Some of them are relatively specific for ARA, such as ACLS4, LPIAT1, and LPCAT3. The polymorphisms and variations in their expression levels can affect ARA deposition in the different tissues, including the brain, despite identical ARA content in the food. Nutritional recommendations regarding the ARA intake must be based on the knowledge of such variations.

ARA’s effects on brain signaling functions depend on the fact that free ARA is released from membrane phospholipid by cPLA_2_α or secretory PLA_2_. The turnover of ARA in brain membrane phospholipids is involved bipolar disorder. The downregulation of the turnover correlates with the reduced expression of ARA-selective cPLA2 or acyl-CoA synthetase and COX-2 [244]. 

It is not known whether higher ARA amounts in specific phospholipid classes in the brain favor ARA released by these PLA_2_ and lead to higher free ARA levels in neuronal or glial cells. Further studies are required to determine the effect of ARA brain uptake on its release and conversion into pro- or anti-inflammatory derivatives.

While the competition between n-6 and n-3 fatty acids and their precursors for the desaturases, COXs, and LOXs are known, little is known on the competition between ARA and DHA in incorporation in the brain. Mfsd2A transports DHA to the brain in the form of DHA-LPC [74], but brain ARA content was increased by 30% in Mfsd2A knockout mice, whereas that of DHA was reduced by 58%. These fatty acid changes were accompanied by neuronal loss in the hippocampus and cerebellum with severe cognitive deficits and anxiety. Therefore, the deposition of ARA to the brain is not altered by the reduced expression of Mfsd2A. However, under this circumstance, ARA replaced DHA in phospholipids but could not play the roles of DHA in neuronal functions [74]. 

FABPs also have different affinities for ARA and DHA. FABP5 and FABP7 are more selective for DHA, whereas FABP3 binds ARA with much higher affinity [263,264]. Regarding fatty acid metabolizing enzymes, DHA is preferentially used as a substrate by ACSL6 [265] and calcium-independent group VI PLA_2_ [266], whereas ACSL4 and group IV cPLA_2_ used ARA as the preferred substrate.

ARA and DHA are preferentially incorporated in different lipid fractions of the brain regions, suggesting a differential role of these lipid biomolecules. ARA is concentrated into brain inositol phospholipid (PL), ethanolamine plasmalogens lipid fractions, while DHA is accumulated in serine and ethanolamine PL fractions of the synaptic membranes. ARA-rich PL is more enriched in white matter (myelinated regions) than grey matter, and there are regional differences in ARA and DHA in the brain. The DHA, ARA, and adrenic acid (22:4n-6) are the most abundant PUFAs in the brain. The preterm infants had less DHA and a lower DHA/ARA ratio in both the brain and the retina than term infants. [267] The mean proportions of ARA were higher in the early placenta than term, while its immediate precursor dihomo-gamma-linolenic acid,20:3n-6 (DGLA) was higher in term than the early placenta. The increased presence of ARA during early placentation supports its organogenesis and vascularization activities. In contrast, the enhanced proportions of DGLA favors the optimal blood flow to sustain fetal growth by their vasorelaxant and anti-platelet effects of PGE_1_-like activities [268,269]. ARA and DHA are required to replenish brain injury, vascular regulation, and brain development in preterm babies. ARA is needed to support endothelial cells during brain injury, while DHA is required to support membrane fluidity for the receptors and the linear growth and network of neuronal cells. The neuroprotective actions of DHA in preterm are manifested with anti-inflammatory initiation and resolutions by its own and its signaling lipid mediators in particular. ARA and DHA produced different sets of pro-resolving lipid bioactive. Lipoxins are derived from ARA, while resolvin (neuro) protectin and maresins are derived from EPA and DHA. LXA4 receptor staining in the brain indicates that lipoxinA4 lowers neuroinflammation and brain edema during brain injury [270]. The anti-inflammatory lipoxinA4 acts as an endogenous allosteric modulator of the cannabinoid receptor [271]. Thus, in contrast to PGE2, which shows pro-inflammatory actions, lipoxins offer an inflammation resolution process.

## 10. Conclusions

The essentiality of both DHA and ARA is known by the fact that these fatty acids make up 20% of the brain’s fatty acids. The mother is required to supply these two fatty acids preferentially during the critical period of brain development via the placenta in utero and breastfeeding postnatally. However, the data on maternal and infant nutritional intakes are not yet consistent, despite the fact that DHA’s impact on brain development and function has been investigated extensively for the last two decades. Due to some compounded variables and conflicting reports on the association with LCPUFA supplementation and cognitive development, it imperative to determine optimum DHA doses in the presence of ARA for optimum brain development in infants [272,273,274]. 

The mean concentrations of DHA and ARA in breast milk can vary based on the maternal diet [260]. The group with the highest levels of DHA showed decreased ARA levels in two brain areas, suggesting the competition of DHA with ARA. Still, the absolute levels of ARA and DHA could be more important than their ratio, in particular with the preterm infant. ARA supports the first year of developmental growth as the conditional deficiency of ARA is established in preterm infants [60]. 

The optimal balance of DHA and ARA intake during infancy is still unknown, but the current best practice advises that the amount of DHA in infant formula should not exceed the amount of ARA [275]. 

The high doses of DHA in formula may suppress the benefits of ARA. The preterm infant who received formula with the n-6/n-3 ratio of 2:1 showed a higher level of LCPUFAs and improved psychomotor development than the n-6/n-3 ratio of the 1:1 group, suggesting an ARA and DHA ratio of 2:1 in the formula for the very preterm infant [276]. Thus, the common requirement for nutritional supports needs both ARA and DHA for vascular and neuronal development, in particular with pregnancies where the supply of these nutrients halted prematurely [277]. The supplementation of DHA:ARA in infant formulas ranges from 1:1 to 1:2. The present recommendations with DHA and ARA levels are that 0.2% to 0.4% and 0.35% to 0.7% of total fatty acids are appropriate [278]. The formula should reflect human milk composition for optimal neurocognitive benefits. Both ARA and DHA are necessary for fetal neurodevelopment, and a deficiency in one may compromise growth. However, further works are required to understand the complementary roles of ARA and DHA in neurodevelopment.

## Figures and Tables

**Figure 1 nutrients-13-02061-f001:**
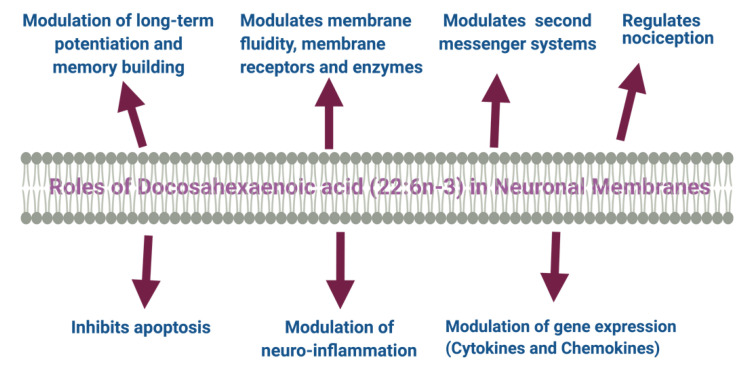
DHA modulates several aspects of structural and functional activities of neuronal membrane.

**Figure 2 nutrients-13-02061-f002:**
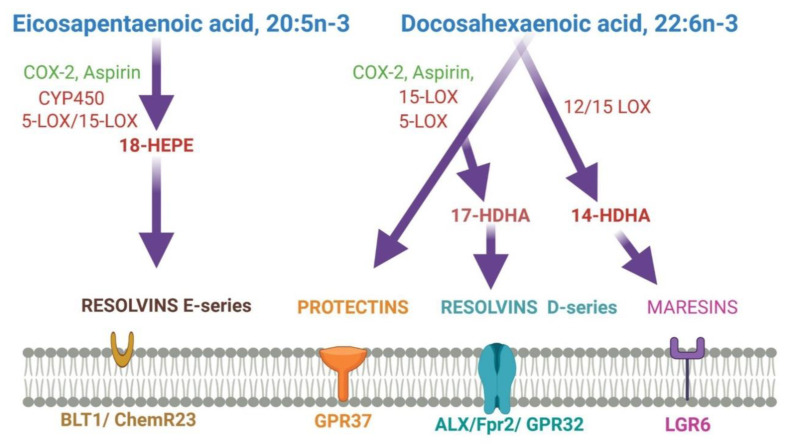
Metabolites of EPA and DHA, and their membrane receptors.

**Figure 3 nutrients-13-02061-f003:**
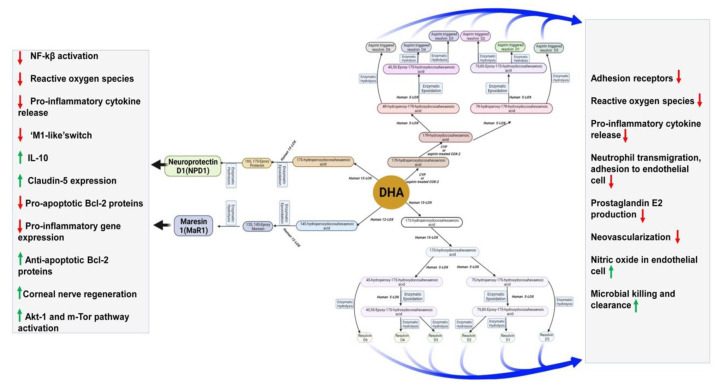
DHA metabolites formation and function in the brain.

**Figure 4 nutrients-13-02061-f004:**
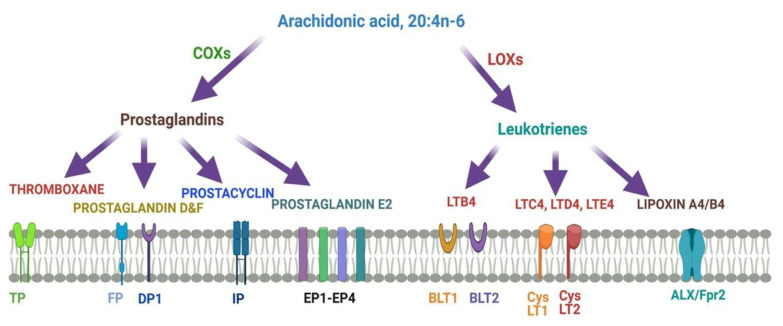
Metabolites of ARA and their receptors.

**Table 1 nutrients-13-02061-t001:** Functions of DHA and ARA metabolites.

Metabolites	Name	Biological Effects
DHA Metabolites	Maresins	Resolution of inflammation, wound healing, analgesic effects
	Protectins	Resolution of inflammation, neuroprotection
	Resolvins	Resolution of inflammation and wound healing
	Electrophilic oxo-derivatives (EFOX) of DHA	Anti-inflammatory, anti-proliferative effects
	Epoxides	Anti-hypertensive, analgesic actions
	Neuroprostanes	Cardio-protection, wound healing
DHA conjugates	Ethanolamines and glycerol esters	Neural development, immunomodulation, metabolic effects
	Branched fatty acid esters of hydroxy fatty acids (FAHFA)	Immuno-modulation, resolution of inflammation
	N-acyl amides	Metabolic regulation, neuroprotection, neurotransmission
ARA metabolites	Lipoxins A4	Lowers neuroinflammation by inhibiting microglial activation
	Lipoxins B4	Promotes neuroprotection from acute and chronic injuries

**Table 2 nutrients-13-02061-t002:** Various clinical studies of DHA supplementation in mothers and infants about brain function.

Study Name	Experimental Setting	Observed Outcome
The Kansas University DHA outcome study (KUDOS) clinical trial	Cognitive and behavioral development	Improvement of visual attention among infants has been observed to reduce the preterm birth risk [234].
Effect of DHA supplementation vs. placebo on developmental outcomes of toddlers born preterm	Developmental outcomes of toddlers	Daily supplementation of DHA did not improve cognitive function and may adversely affect language development and effortful control in specific subgroups of children [235].
Effect of DHA supplementation during pregnancy on maternal depression and neurodevelopment of young children	Neurodevelopmental outcome of children	DHA supplementation during pregnancy did not reduce postpartum depression in mothers, neither did it improve cognitive and language development in their offspring during early childhood [236].
Neurodevelopmental outcomes of preterm infants fed high-amount DHA	Neurodevelopment at 18 months of age	Bayley mental development index scores of preterm infants overall born earlier than 33 weeks were not affected but improved the girls’ Bayley mental development index scores.
Neurodevelopmental outcomes at 7 years corrected age in preterm infants who were fed high-dose DHA to term equivalent	Cognitive outcome detected at 18 months age	No evidence of benefit [237].
Feeding preterm infant milk with a higher dose of DHA than that used in current practice	Language or behavior in early childhood	No clinically meaningful change to language development or behavior were observed when assessed in early childhood [238].

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
