# Peer review of "Maternal Supply of Both Arachidonic and Docosahexaenoic Acids Is Required for Optimal Neurodevelopment"

_nutrients, 2021, doi:10.3390/nu13062061_

Round 1

Reviewer 1 Report

The title does not give a clear indication of the topic and research question being addressed.

Insufficient explanation of the rationale for the study, that makes the manuscript difficult to read and understand.

The manuscript contains a profusion of repetitive elements.

Insufficient description of methods.

The number of references is large and comprehensive, although the main conclusions drawn from results are not clear. The format of the manuscript is not coherent.

In relation to the formal academic writing there are too many short simple sentences that are disconnected, and also numerous highlighted words and letters of different colours and different sections of the manuscript.

Author Response

Reviewer-1

The title does not give a clear indication of the topic and research question being addressed.

 Rebuttal: The title is now justified with an elaborate discussion on ARA and its interplay with DHA on brain development.

Insufficient explanation of the rationale for the study, that makes the manuscript difficult to read and understand.

Rebuttal: This has now been rephrased and revised to make a better rationale for this review manuscript. In recent years, there has been much discussion about the need for the addition of ARA and DHA in infant formula. We think it is essential to highlight the importance of ARA, in addition to DHA, in early brain development about fetal and postnatal brain development and the maternal supply of ARA and DHA. The rationale is now vividly explained where importance of ARA is emphasized.

The manuscript contains a profusion of repetitive elements.

Rebuttal: We appreciate the reviewer’s comments, these issues are now addressed in the revised manuscript.

Insufficient description of methods.

Rebuttal:  This has been addressed now. A literature search was performed on the PubMed database using search terms such as n-3, n-6 fatty acids, DHA, ARA, brain development, infant, fetal, breastfeeding, nutrients deficiency, supplementation. All types of articles related to humans only were included for evaluation. Articles for which full text was not available and papers that were not in English were excluded. The articles retrieved in the first round of searches identified additional references by a manual search among the cited references.  

The number of references is extensive and comprehensive, although the main conclusions drawn from results are not clear. The format of the manuscript is not coherent.

Rebuttal:  We appreciate your insight and comments. We have now deleted around 25 or more references. We have also extensively modified and organized the manuscript in order to arrive at the conclusions. The format of the manuscript is now modified.

In relation to formal academic writing there are too many short simple sentences that are disconnected, and also numerous highlighted words and letters of different colours and different sections of the manuscript.

Rebuttal:  These have been taken care of in the revised manuscript

Reviewer 2 Report

Dear Authors,

Please find enclosed the pdf file with the Reviewer comments. 

Best Regards.

Author Response

Reviewer-2

As the title suggests and the main goal presented in the abstract and at the end of introduction, this review describes the importance of maternal-fetal preferential transfer of docosahexaenoic acid (DHA) and arachidonic acid (ARA), two main long chain polyunsaturated fatty acids, to support the infant's optimal brain development and growth and functional roles in the brain.

In general, the review is well written and reports many interesting information but some parts are too long with some information not really new especially for DHA or not essential for the aim of this review. Indeed, several parts in the different paragraphs describe the effect of DHA in brain function in teenagers and adults, which is far from the maternal DHA transfer to the fetus and to the baby during the pregnancy and breastfeeding. The authors should focus more on the balance between DHA and ARA transfer as a major condition to maintain a positive effect of LCPUFA in the neurodevelopment.

I regret that there is only one paragraph (the last one) addressing the ARA effect in the brain. I did not review the Figures due to no access to them!

Rebuttal:  We appreciate your constructive, minute, and detailed feedback on our manuscript. At first, we sincerely thank you for reviewing our manuscript thoroughly. Each of your concerns is addressed pointwise and incorporated changes within the text.

In order to explain the mechanism of DHA’s neurodevelopment actions, adult and teenager instances were considered in some cases. Because of existing information on DHA, we have now revised the draft extensively. A section is added on the roles of ARA  in brain development.

We attempted to highlight the reports those focused on the balance between DHA and ARA transfer. However, data are still emerging on ARA effects in the brain, particularly cases where gestational windows are terminated early.

In Abbreviations, please add the definition of RCT

Rebuttal:  It is now added

  1. Introduction (p2):

General findings:

The introduction is well written but it is focused on the n-3 LCPUFA and especially the DHA effect in neurodevelopment and does not report studies on the ARA effect. All the n-6 LCPUFA effect including those of ARA are missing and should be reported as DHA is.

Rebuttal:  The importance of ARA and DHA has now been added in the introduction section of page 2.

Minor corrections
-lines 42-43 The authors write “Since de novo synthesis of DHA is low in a growing fetus, ...” This report needs a reference.

Rebuttal:  The reference is added

-line 51 “Multiple trials have found that higher plasma or erythrocyte DHA levels positively correlate with... “ Is it in the plasma of mother or fetus? Please add his precision.

Rebuttal:  It is now revised

-lines 60-62 “In the central nervous system (CNS), the proportion of DHA with other membrane fatty acids (FA) increases as the brain size increases.” This sentence underlines that other membrane FA may be involved in the brain growth which is an important point. The authors should add reference and maybe develop a little bit this argue.

Rebuttal:  The statement indicates the content of DHA proportion with other fatty acids increases with brain size. Due to the lack of scope in this article, details of different fatty acids are not elaborated.

-lines 80-81 Similar remarks here, the authors suggest that “the variation in the fatty acids composition of mother’s milk...” could be interesting to develop more.

Rebuttal:  We agree with the reviewer this could be interesting to pursue in the future as a follow-up study.

  1. Maternal delivery of DHA to the developing brain (p3)

General comment: Maternal delivery of ARA to the developing brain is not addressed in this review. Why did the authors choose to describe the maternal delivery of DHA and not ARA? This paragraph should present both DHA and ARA.

Rebuttal:  Maternal delivery of ARA and its mechanisms are introduced in a separate paragraph later. The importance of DHA has been described with concurrence with ARA as far as possible. However, the introduction to DHA on neurodevelopment is necessary to understand the ARA effects since later is not being viewed as independent effects.

-lines 117-119 the sentence starting by “DHA incorporation in the neuronal membrane in early life...I guess for the authors “early life” refers to the first 1000 days (DOhAD concept) including fetal life, breastfeeding and child growth to 2 years old. So, the two references 34 and 35 correspond solely to fetal life instead of Breastfeeding and endogenous synthesis of DHA by the liver of the baby. Please replace those references or add appropriate literature for Breastfeeding and postnatal child growth.

Rebuttal:  The statement includes the importance of endogenous DHA levels due to breastfeeding on early life, including feto-placental and infant neurodevelopment. These two reviews cover several aspects of these issues. In addition to those, one more specific reference has been added in that section.

-line 122, About ref 37, there is an error in the title of ref 37 reported line 817 p16. The title of this article is “Docosahexaenoic acid” instead of “The DHA content...to signals”

Rebuttal:  This is now revised

lines 132, ref 44 concerns the feeding of preterm neonates immediately after early birth and during the hospital period. Not sure that study in preterm babies is comparable to breastfed infants born at term.

Rebuttal:  The statement was made a combined implication from both references 43 and 44. In which both term infants and premature infants were studied independently.

-lines 133-137. The authors write: “the preferential postnatal accumulation of LCPUFAs in infant brain tissues is mediated via breastmilk”. And add in line 141; “Breastfeeding is vital for an infant’s neurodevelopment.” but because not all women breastfeed, what about non- breastfed children? Do they all develop neurodevelopmental disorders? The information should be discussed as well.

Rebuttal:  The development of neurodevelopment disorders in non-breastfed children can be an interesting area to pursue further. However, there is little scope in the present context of the subjects. 

  1. The fatty acid uptake system of the brain (p4)

-lines 163-164. The sentence is confusing. The authors report in this sentence that “the passive diffusion of non-esterified DHA or ARA .... seems the major entries...of these fatty acids within the BBB. However, all the references used here (ref 55-61) mention the lysophosphatidylcholine (LPC) as the main lipid form to transfer DHA to the brain within the BBB mediated by the LPC transporter, MFSD2a. So DHA is mainly transfer to the brain as esterified in LPC over the non-esterified form. Please modify the sentence and maybe reduce the number of references.

Rebuttal:  The recent evidence indicates LPC-DHA carrier mediated transfer is predominant over passive diffusion with reference to crossing BBB. We have revised this section.

-lines 173-175. Please add a reference at the end of this sentence.

-lines 175-177. Please add a reference at the end of this sentence as well.

-lines 180-182. A reference is missing here.

All these sentences are intermingled within the context of the newly discovered DHA transporter in the brain. Thus, the references are also mixed in this section.

-lines 194-195. In the sentence beginning by “FABP3 is expressed..., please delete the word “levels” after “expression”.

Rebuttal:  This is now revised

-lines 206-208. The references 77 and 78 present the general FABP affinity for different types of FA. Is it working for brain cells?

Rebuttal:  These also include brain cells

-Lines from 191 to 214 this part could be shortened to highlight solely the role of FABPs in binding for LCPUFA, in particular ARA and DHA.  Paragraph from lines 251 to 236 is very interesting but again it would be more valuable for the aim of this review to show a link (If it is known) between a potential defect of FABP expression and DHA or ARA availability in brain disorder.

Rebuttal:  FABPs as carriers of LCPUFAs are emerging since tissue and stage-specific expression of FABP isoforms are reported. The importance of preferential binding of FABPs of ARA and DHA is also highlighted in another section.

  1. Structural and functional roles of DHA in the human brain (p5)

References 90 and 98 are the same reference: Calderon, F.; Kim, H.Y. Docosahexaenoic acid promotes neurite growth in hippocampal neurons. 969 Journal of Neurochemistry 2004, 10.1111/j.1471-4159.2004.02520.x, 970 doi:10.1111/j.1471-4159.2004.02520.x. Could you please check it?

Rebuttal:  This is now revised

-Lines 260-262: Several studies emphasized the ... Please add references here.
-Line 265: The authors write: “The ALA restricted diet .... due to lack of DHA”. Is it “lack of DHA synthesis from ALA? Please clarify that point and please check the reference 98.

Rebuttal:  The statement in line 260 is revised now as a general comment. In rats, generally, ALA deficiency reflects DHA deficiency. Reference 98 is now revised.

-Line 278: Please modify the sentence by keeping only the second part. “DHA and EPA stimulate the angiogenesis in the placenta (38)”.

Rebuttal:  The reference is appropriate for the second part of the statement, but their contrasting effects in cancer is required to mention here.

General point, paragraphs 3 and 4 should be compiled and shortened.

Rebuttal:  These are now revised

  1. Roles of DHA and its metabolites in the brain (p6)

-Line 298-300: Please add reference about oxylipins that regulate ...

Rebuttal:  Reference is now added

-Line 303-305: Is it in the brain? What about the activity of desaturase and elongase enzymes involved in the essential FA conversion to DHA or ARA?

Rebuttal:  The activities of desaturase and elongase are involved stepwise in converting n-3 PUFA in to long chain n-3 PUFA. We skipped these as it is already known and established conclusively.

-Line 305-308: “DHA levels are decreased...” These two sentences should be moved in the paragraph 6 (6. DHA deficiency and human brain function) which talk about DHA deficiency.
-Line 310-311: “DHA stimulates membrane-associated G-protein...” Is this in the brain?

Rebuttal:  This section has been described in the context of DHA’s mechanism of action in general and its possible implication in the brain. These mechanistic data are derived from deficient animal models in brains.

  1. DHA deficiency and human brain function (p8)

In this paragraph, the authors should specify when the deficiency of DHA come from the mother (pregnancy and lactation) or later in life. For example, lines from 413 to 415, “Neurocognitive functional insufficiency in young adults.... associated with DHA deficiency”

Rebuttal:  The DHA deficiency in utero is applicable everywhere. The DHA deficiency carries forward if not intervened adequately. The section heading is now revised.

The separation of the two distinct periods (first 1000 days and later in life) could reinforce the crucial role of DHA in early brain development and in brain function along the life. However, the part relative to later in life (teenager and adult) has to be shortened to match with the title and the aim of review which is focused on neurodevelopmental period.

Rebuttal:  The cause and consequences of DHA deficiency are lifelong. It is sometimes difficult to trace it later unless the mechanisms of DHA’s neurodevelopment actions are monitored in adult and teenager stages. Therefore, DHA’s function on neurodevelopment must be viewed holistically that initiated in utero.

-Lines 415-417 “DHA deficiency in the third trimester ...to insufficient maternal consumption of n-3 fatty acids”. This sentence needs a reference in preterm neonates.

Rebuttal:  Since it is fact now, a separate citation was not used.

-The lines from 438 to 449 discussed about DHA effect on serotonin in the brain and association with schizophrenia. The associated references are in adults. It would be interesting to show the effect of maternal DHA deficiency on serotonin metabolism in fetal brain or offspring brain. Are there any pre-clinical studies on that?

Rebuttal:  Clinical data on maternal DHA deficiency on serotonin metabolism in the fetal brain or offspring brain is not available.

  1. Can DHA supplementation improve brain function of infants: results of clinical trials (p9)

In this paragraph, it could be useful to add when the DHA supplementation was giving to the mother during the pregnancy. First, second or third trimester and during the lactation?

Rebuttal:  Several clinical trials were performed on DHA supplementation at different time points but not before 14 weeks of gestations.

Line 479. This sentence needs a reference.

Rebuttal:  This is a general assertive statement from several previous works, therefore not explicitly cited.

  1. Roles of arachidonic acid in brain development and function (p 10)

Why some parts are underlined only in this last paragraph?

Rebuttal:  Possible typos. Now it is revised.

-Lines 507-508: “ARA may be required in a higher amount to support growth-promoting placental activities and production of eicosanoids”
Do the authors mean that uptake of ARA “to support growth-promoting placental activities and production of eicosanoids” is higher than those of EPA and DHA, only at early pregnancy? Please clarify?

Rebuttal:  In vitro study involving invasive trophoblast mimicked the first trimester showed higher uptake of ARA in the placenta. However, clinical data is required to confirm this finding.

-Lines 512-514: Please add a reference

Rebuttal:  It is an assertion for pregnant women therefore references are not added. Female is replaced with pregnant women.

-Lines 525-526: Please add a reference about the low conversion of LA into ARA in the brain.

Rebuttal: The brain is not metabolically active like the liver in converting LA to ARA. Thus, preformed ARA and DHA are preferred to the brain. It is an assertive statement.

-Lines 527: Please move the reference number at the end of the sentence.

Rebuttal:  Insertion of reference to the particular place will help reader to track the specific fact

-Lines 551-554: These sentence needs a reference.

Rebuttal:  This section has been revised now.

-Lines 562-563: “The authors write that ARA has pleiotropic effects on brain disease...” such as? Please give an example showing the pleiotropic effects of ARA.

Rebuttal:  A new paragraph has been added in this section on the different functions of ARA.

-Lines 577-578: “A diet containing 2% ARA for 21 weeks...” Please add the reference

Rebuttal:  A reference is added

-Lines 646-647: Please add the reference in preterm babies.

This section has been revised now.

  1. Conclusions

-Lines 673-674: Epidemiological studies report a ...Please the references concerning those studies

Rebuttal:  This statement has been revised now.

-Line 678. Please correct the error “The ARARA” by the ARA-mediated processes: -Line 681. Please remove the word ‘brain” before DHA.
The reference number 273 displays an error in the title. Please check it.

Rebuttal:  These typos are revised now. The references are now revised after updating the endnote.

Reviewer 3 Report

 This  is very interesting manuscript on the relative importance of DHA/AA in the neurological development of children.
My comments
1/ It is a very extensive paper on animals mainly and little human data and little literature on child development.
The authors of this paper's documentation on arachidonic acid (AA) was not convincing on the relationship between AA and neurodevelopment in children.
While DHA plays a role in animal neurodevelopment, animal brain data, and adult human diseases such as Alzheimer's and Parkinson's, it is ethically difficult to obtain brain tissue from infants, and we do not have much data on infant neurodevelopment in this article.
But the well-sourced cellular and clinical data from adults and children from the extensive literature review (272 references) are convincing of the mostly biological effects of DHA's neurological benefit.
2/ Arachidonic effects are more difficult to prove, but as all lipid researchers suggest, DHA and AA contribute to infant neurodevelopment.
3/ The relative n-6/n-3 ratio of precursors is not detailed and the recommendation of the French Nutrition Committee is close to 6.
4/ The relative AA/DHA ratio is at least 1 or 2 as suggested by the author, even in case of nutrition committee advice.
When we read the world levels of Brenna in 64 countries, DHA could vary according to the fatty fish consumed from 0.07% in poor African countries, 0.12% in the USA and Canada, while in Asia, DHA in Japan is 1% and in maritime China 2.78%.
AA varies less between 0.50% and 1.2% in maritime China and 0.4% in Japan without IQ deficiency in Japan.
So it is true that ARA is always higher than DHA but in Japan. And China or all peoples who consume fish.  
In France we see from 1997 to 2014 a decrease in linoleic acid and an increase in linolenic acid and an increase in DHA from 0.24 to 0.29. 

Author Response

Reviewer-3

This  is very interesting manuscript on the relative importance of DHA/AA in the neurological development of children.

Rebuttal:  Thank you for the positive comments on the manuscript. We appreciate it very much,

My comments
1/ It is a very extensive paper on animals mainly and little human data, and little literature on child development.

Rebuttal:  We appreciate the comments, although we disagree to some extent. Some model studies are included in the manuscript to describe nutrient actions in the developing brain under deficiency conditions. We think this is an important insight to understand the functions of fatty acid carriers in the developmental brain.

The manuscript was primarily included as much as data available on the infant, children, and fetal brain development.

The authors of this paper's documentation on arachidonic acid (AA) was not convincing on the relationship between AA and neurodevelopment in children.v But the well-sourced cellular and clinical data from adults and children from the extensive literature review (272 references) are convincing of the mostly biological effects of DHA's neurological benefit.

Rebuttal:  We agree that AA (or ARA) data on neurodevelopment is not as extensive and convincing as DHA. Nevertheless, a comparative review article is required for future studies, as these two LCPUFAs are involved in the structure and function of the brain. However, we have added new info on ARA.

ARA is involved in cell division and signaling during brain growth and development (239).  In addition, ARA mediates neuronal firing (240), signaling(241)and long-term potentiation(242). ARA maintains membrane order and hippocampal plasticity(243). ARA also protects the brain against oxidative stress in the hippocampus region via PPARγ and synthesizes new proteins (244). Released unesterified ARA activates protein kinases and ion channels, inhibits neurotransmitter uptake, and enhances synaptic transmission, and modulates neuronal excitability (239). As ARA mediates intracellular signaling, the concentration of free ARA must be maintained at precise levels within the cells.  

ARA also is responsible for the activation of syntaxin-3 (STX-3), a plasma membrane protein involved in the growth and repair of neurites(245).  STX-3 serves as a single effector molecule and direct target for ARA(245). Neurite growth closely correlates with the ability of ARA to activate STX-3 in membrane expansion at growth cones(245). The growth of neurite processes from the cell body is a critical step in neuronal development

ARA also enhances the engagement of STX-3 with the fusogenic soluble N-ethylmaleimide-sensitive factor attachment protein receptors (SNARE complex), proteins that form a ternary complex that drives exocytosis(246). In the brain, at the neuromuscular junction, and in endocrine organs, a set of three SNARE proteins has a primary role in producing fusion of vesicular and plasma membranes. The formation of this SNARE complex drives membrane fusion, leading to the release of vesicular cargo into the extracellular spaces(246). Darios and colleagues (246)report that α-synuclein, a synaptic modulatory protein implicated in the development of Parkinson's disease, can sequester ARA and block the SNARE complex's activation. This finding underlines the importance of ARA for the regulation of synaptic transmission and transport. The biochemical functions of ARA demonstrate its importance for cell signaling, trafficking, and regulation of spatial-temporal interactions between cellular structures. However, the formula with the highest level of DHA significantly reduced ARA levels in two areas of the brain (superior colliculus and globus pallidus), indicating its competition with ARA and the importance of a proper balance of DHA to ARA

2/ Arachidonic effects are more difficult to prove, but as all lipid researchers suggest, DHA and AA contribute to infant neurodevelopment.

Rebuttal:  Agree. However, we collated all the available information on the contributions of AA on human brain development

3/ The relative n-6/n-3 ratio of precursors is not detailed and the recommendation of the French Nutrition Committee is close to 6.
4/ The relative AA/DHA ratio is at least 1 or 2 as suggested by the author, even in case of nutrition committee advice.

Rebuttal:  We appreciate this, however, the then-6/n-3 ratio of the precursors not detailed here, as the conversion rate of the precursors to ARA and DHA is highly compounded. Usually, baby food is supplemented with ARA and DHA. Therefore, we restricted our viewpoints on the ARA and DHA. 

When we read the world levels of Brenna in 64 countries, DHA could vary according to the fatty fish consumed from 0.07% in poor African countries, 0.12% in the USA and Canada, while in Asia, DHA in Japan is 1% and in maritime China 2.78%.
AA varies less between 0.50% and 1.2% in maritime China and 0.4% in Japan without IQ deficiency in Japan.
So it is true that ARA is always higher than DHA, but in Japan. And China or all peoples who consume fish.  
In France we see from 1997 to 2014 a decrease in linoleic acid and an increase in linolenic acid, and an increase in DHA from 0.24 to 0.29. 

Rebuttal :  Yes, this reflects worldwide AA and DHA distribution in human breastmilk. This is one of the rationales along with the presence of AA and signaling in the human brain underscores the importance of AA in the human brain

Round 2

Reviewer 1 Report

I believe that the manuscript presents issues that need to be corrected for publication:

  1. There are too many short simple sentences that are disconnected.
  2. There are concepts that are repeated throughout the manuscript on numerous occasions, such as roles of DHA in functional brain development of the fetus, relationship between DHA and diseases such as Alzheimer's disease, Parkinson's disease, mood disorder, cognitive function…Please review the text, the reiteration of content is not appropriate for a scientific manuscript.
  3. There is no clear answer to the question that gives the title to the article.

Minor revision

Line 18. Arachidonic acid,20:4n-6; Brain; Docosahexaenoic acid; 22:6n-3; Check commas and spaces

Line 36. cosahexaenoic acid,22:6n-3; EPA: Eicosapentaenoic acid,20:5n-3; Check commas and spaces

Lines 44-46. Development/developmental is repeated six times in the same paragraph.

Line 48 acid,22:6n-3 (DHA), and arachidonic acid,20:4n-6 (ARA). Check commas and spaces

Lines 126-129. Development is repeated three times in the same paragraph.

Lines 131-132 "DHA incorporation in the neuronal membrane in early fetal life solely depends on placental transfer [11], breastfeeding, and endogenous synthesis of DHA". Fetuses do not breastfeed

153-159. Different font color

Line 184 but not unesterified DHA[12,56-58].  References are listed in the subindex

Lines 215 and 217 "c>16". You must write c in capital letters

Lines 164 function., Comma left out

Lines 272-274. Different font color

Figure 3. The resolution of figure three is not good

Table 1. Table 1 is out of order in relation to the text.

Table 2. Table 2 is out of order in relation to the text

Line 456. results in downregulation. Remove bold words

Author Response

I believe that the manuscript presents issues that need to be corrected for publication:

  1. There are too many short simple sentences that are disconnected.

These issues are addressed now  in the revised manuscript

  1. There are concepts that are repeated throughout the manuscript on numerous occasions, such as roles of DHA in functional brain development of the fetus, relationship between DHA and diseases such as Alzheimer's disease, Parkinson's disease, mood disorder, cognitive function…Please review the text, the reiteration of content is not appropriate for a scientific manuscript.

We appreciate the comments. These are now taken care of in the revised manuscript

  1. There is no clear answer to the question that gives the title to the article.

The purpose of this review is to describe the importance of arachidonic acid (ARA) and its relationship with vital docosahexaenoic acid (DHA) for the brain development of fetal/infants. ARA is indispensable for brain growth, where it plays a vital role in cell division and signaling. The brain in mammals consists of 60% fat, which requires DHA and ARA to grow and function. ARA is one of the most abundant fatty acids in the brain, and compared with DHA;.

ARA has very different biological functions than DHA. In comparison, DHA controls signaling membranes in the photoreceptor, brain, and nervous system. ARA is vital for brain growth during gestation and early infancy, where it plays a critical role in cell division and signaling

The benefits of ARA plus DHA supplementation cannot be ascribed to DHA alone. The combination of ARA and DHA has shown benefits for cognitive development, visual function, well beyond the period of supplementation and into early childhood.

Without such an assessment, and in light of the universal presence of ARA in human milk and the numerous essential ARA functions for cell structure and function, the most reasonable approach is to include ARA in DHA-containing infant formulas to promote optimal infant growth and development.

Since this review focused on ARA's essentiality for brain growth and development with reference to DHA, we think this title is the most suitable one.

Minor revision

Line 18. Arachidonic acid,20:4n-6; Brain; Docosahexaenoic acid; 22:6n-3; Check commas and spaces

Now it is revised

Line 36. Docosahexaenoic acid,22:6n-3; EPA: Eicosapentaenoic acid,20:5n-3; Check commas and spaces

Now it is revised

Lines 44-46. Development/developmental is repeated six times in the same paragraph.

The term has been replaced with appropriate alternate wherever it is possible

Line 48 acid,22:6n-3 (DHA), and arachidonic acid,20:4n-6 (ARA). Check commas and spaces

Now it is revised

Lines 126-129. Development is repeated three times in the same paragraph.

The term has been replaced with appropriate alternate wherever it is possible

Lines 131-132 "DHA incorporation in the neuronal membrane in early fetal life solely depends on placental transfer [11], breastfeeding, and endogenous synthesis of DHA". Fetuses do not breastfeed

Now this section is revised

153-159. Different font color

It is not clearly showing in the word processor

Line 184 but not unesterified DHA[12,56-58].  References are listed in the subindex

Now it is revised

Lines 215 and 217 "c>16". You must write c in capital letters

It is done

Lines 164 function., Comma left out

Now it is revised

Lines 272-274. Different font color

It is not clearly showing in the word processor

Figure 3. The resolution of figure three is not good

The figure is large to accommodate different pathways togethers. Therefore, font sizes are smaller.

Table 1. Table 1 is out of order in relation to the text.

We made table in word processor with default size.

Table 2. Table 2 is out of order in relation to the text

We made table in word processor with default size.

Line 456. results in downregulation. Remove bold words

Now it is revised
